# The LCA Commons—How an Open-Source Repository for US Federal Life Cycle Assessment (LCA) Data Products Advances Inter-Agency Coordination

**Ezra Kahn** * , **Erin Antognoli and Peter Arbuckle**

USDA, Agricultural Research Service, National Agricultural Library, 10301 Baltimore Ave., Beltsville, MD 21218, USA; erin.antognoli@usda.gov (E.A.); peter.arbuckle@usda.gov (P.A.)
* Correspondence: ezra.kahn@usda.gov

**Abstract:** Life cycle assessment (LCA) is a flexible and powerful tool for quantifying the total environmental impact of a product or service from cradle-to-grave. The US federal government has developed deep expertise in environmental LCA for a range of applications including policy, regulation, and emerging technologies. LCA professionals from across the government have been coordinating the distributed LCA expertise through a community of practice known as the Federal LCA Commons. The Federal LCA Commons has developed open data infrastructure and workflows to share knowledge and align LCA methods. This data infrastructure is a key component to creating a harmonized network of LCA capacity from across the federal government.

**Keywords:** data science; open science; LCA; MFA; input–output analysis; environmental impact; footprint; FAIR data; government data

## 1. Introduction

The Federal LCA Commons is a US federal government community-of-practice of program managers, researchers, and regulators using life cycle assessment (LCA) in support of several US government agency missions. The community coordinates around the open access and reusability of federally funded and federally produced LCA data, models, and research results. The LCA Commons community makes its work available through an open repository at www.lcacommons.gov. LCA data and models produced by LCA commons participants cover the breadth of government interests, including agriculture, energy, transportation, buildings, construction, defense, human health, and environmental protection. Together, LCA Commons expertise covers a large part of the US economy. The LCA Commons community and open data repository are the foundations of an emerging distributed data development effort within the federal government. The community approach of the LCA Commons makes it flexible and adaptable to the drivers of its members. This approach also creates a unique challenge—how to coordinate disparate research and modeling initiatives into an interoperable data network from the grassroots of government organizations? The LCA Commons address this challenge by prioritizing professional networking, open sharing of information, and development of a vision common to its participants. From this social networking emerges a set of specific requirements for services supporting interoperability of LCA data and models between and across government activities.

In this paper, we will describe: (a) LCA and how the government applies it; (b) how this repository advances the access and reusability to these high value, complex data models; and (c) why the repository and the LCA Commons community-of-practice is necessary.

*1.1. An Introduction to Life Cycle Assessment*

Life cycle assessment (LCA) is an approach for quantifying the environmental impacts of products and services, standardized by the International Organization of Standardization [1]. LCA takes a 'cradle-to-grave' perspective by considering all phases of production, use, and end-of-life. An LCA of a product considers extraction of raw materials, transformation of materials to products, distribution and use by a consumer, and the recycling and disposal of the product. This includes the transportation, the heat and energy inputs, infrastructure, and services, and most importantly the emissions at every life cycle stage. Combining all the emissions into a single inventory, normalized to a functional unit, allows for comparisons with other technologies that perform the same function. Life cycle impact assessment aggregates and characterizes the final emissions profile to potential impacts of interest (greenhouse gas contributions to global warming potential is the best-known example, but eco-toxicity, photochemical smog, and human health effects are also common impacts estimated in LCA). LCA provides a holistic picture of the total emissions and potential impact of an economic activity, including those outside the immediate control or scope of a decision maker. It parses the total emissions down to contributions from individual activities allowing for the identification of 'hotspots'—opportunities to improve efficiency, reduce impact, and improve the overall performance of the system.

In practice, an LCA is produced in four phases: goal and scope definition, life cycle inventory, life cycle impact analysis, and interpretation (see Figure 1). The goal and scope definition establishes the intention of the LCA, the study stakeholders, the processes and technologies of interest, and the expected application of the LCA results. The goal and scope definition sets the requirements for data collection regarding which processes are under consideration, and which flows to and from nature, must be measured to quantify impacts of concern (such as global warming, water quality, human health, etc.). During goal and scope definition, influential values choices are specified according to the intention of the LCA. Examples of value choices include: identifying critical life cycle processes requiring primary data collection, identifying general background data sources such as electricity and waste management models, choosing an impact method to characterize emissions into impacts, setting cut-off criteria for excluding minor or non-influential contributions, and setting rules for aggregation, co-product management, and other LCA specific modeling decisions.

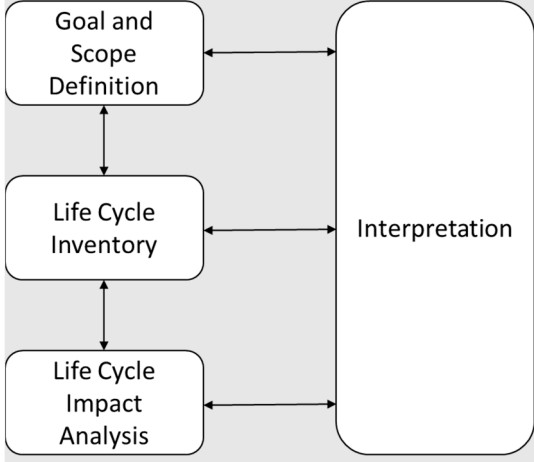

**Figure 1.** The four phases of life cycle assessment.

The requirements outlined in the goal and scope phase directly inform the data collection, organization, and modeling work, known as the life cycle inventory phase. The data intensive nature of the modeling approach often makes the life cycle inventory phase the most expensive part of performing LCA. Primary data for LCA is collected for key processes identified during goal and scope definition. Secondary data in the form of past

LCA work, engineering models, and interviews with subject matter experts is collected to fill out the rest of the supply chain. The modeling decisions and value choices documented during goal and scope definition are implemented, and produce a life cycle inventory (LCI). An LCI is a single list of emissions and quantities representing the entire production life cycle of a product of interest, normalized to a functional unit. The LCI illustrates the component emissions that contribute to impacts (for example: the $CO_2$, methane, and nitrous oxides inventory all contribute to global warming potential).

The LCIA result is the primary quantitative information product of an LCA study. The LCIA result is produced by characterizing the LCI by a life cycle impact assessment method (LCIA) into an impact of interest. For example, GWP is typically reported as $kgCO_2$ equiv., where $CH_4 = 28$ $CO_2^{equiv}$ and $NOx = 265$ $CO_2^{equiv}$ [2]. The data collection method, how the scoping decisions influenced the LCI model, and impact method used to characterize the LCI heavily influences the LCIA result of a particular product under study. The interpretation phase analyzes the validity of these decisions with respect to the original intention of the LCA and its stakeholders. Reiterating LCAs after the interpretation phase often improves the study's representativeness with regard to its stated goals. This could include improving data quality, collecting more primary data, or modifying the modeling decisions.

The LCA product often takes the form of a written report documenting the decisions and outcomes from the four phases of the LCA study. The LCA often includes a discussion on the environmental performance of the system, recommendations or opportunities for reducing impact, and any tradeoffs between impacts of interest that arise from choices and decisions producers may have. The final LCA report does not typically share the complete LCI or the specific process and modeling information that contributes to the LCI (known as 'LCI data'). However, because the collection and preparation of the LCI data are usually the most expensive part of the study, they can be valuable for reuse as secondary data in other studies. LCI data from previous LCAs of similar systems often provide credible data inputs where suitable according to goal and scope definitions. LCI reuse can reduce costs and increase the comparability of LCA studies covering similar product functions—two drivers for the LCA Commons community.

Preparing LCI data for reuse is an ongoing challenge for the LCA community. The sensitivity of the original goal and scope to the LCI data complicates sharing LCI data for use in new LCA studies. Enough information to determine the suitability of an LCI data set to an application different from its original intention requires more information than LCI or LCA reports typically include. The LCA community currently addresses this challenge through the creation of metadata, the use of structured formats, and information sharing workflows. The LCA Commons makes advances in this space, but a lot remains to be done to achieve what the LCA community calls 'interoperable data'; secondary data repurposed for a new goal and scope.

To date, no official unified US federal government LCA initiative that leverages and co-ordinates the breadth of federal expertise exists, due in part to the organizational structure of the US federal government itself. The federal government is organized vertically with independent departments responsible for specific sectors of American society. Each department and agency operates within the scope of its mission to meet Executive or Legislatively defined program requirements using deep expertise in subject matter specific to the agency. This structure supports good communication up and down organizational leadership and the flexibility to adapt to change. The interdisciplinary scope of LCA provides a framework for horizontal interagency coordination by linking together expertise from different parts of government into a holistic decision support tool.

### 1.2. LCA Applications and the US Federal Government

LCA has developed across the federal government independently, with the greatest investments happening since the mid-1990s. The best-known examples of federal LCA work consist of modeling tools and datasets developed at the US Environmental Protection

Agency (EPA), Department of Energy (DOE), and National Institute of Standards and Technology (NIST). In the 1990s, the US EPA developed TRACI, a standardized set of impact characterization factors for the United States [3]. The EPA continues to maintain and update TRACI, providing one of the necessary pieces of the puzzle to consistently use LCA in policy and regulatory contexts. The DOE Argonne National Laboratory has been developing the Greenhouse Gases, Regulated Emissions, and Energy Use in Transportation (GREET) tool since 1996. GREET provides an integrated model of on-road transportation, including conventional and advanced technologies. A range of applications use this model—for example the California Air Resources Board uses GREET as a calculator for managing the California Low Carbon Fuels Standard regulation [4]. The National Renewable Energy Laboratory has maintained the US LCI since 2001. The US LCI makes industry specific and general US background data available through an open repository [5]. The National Institute of Standards and Technology (NIST) has been developing the Building for Environmental and Economic Sustainability (BEES) tool, an LCA tool for building materials used by the Green Building Council for LEED certification. The National Energy Technology Laboratory produces LCA work that serves as the foundation for government programs such as the federal Renewable Fuels Standard (RFS2) and the Carbon Dioxide Utilization Program (CO2U).

The GREET tool from Argonne National Laboratory clearly illustrates the many sectors of industry and government that can touch a single LCA application. On-road transportation is the primary scope of GREET, which has two main components: vehicle manufacture and fuel pathways. The construction of vehicles requires metals and other critical minerals, rubber and plastics, glass, and fabrics, as well as transportation, manufacturing, infrastructure, and energy. The development of conventional fuel pathways covers the life cycle of fossil fuel extraction, transport, refining, and combustion. Alternative fuels require data representing new feedstocks—such as corn, soybeans, or miscanthus—and energy carriers such as lithium-ion batteries and hydrogen fuel cells cover a range of specialized materials. Finally, the characterization of emissions into impacts of interest require knowledge in risk assessment and toxicology. As such, no one person or organization has all the knowledge necessary to produce LCA results. Instead, LCA relies on expertise drawn from a range of sources, literature, databases, and tools (both public and proprietary), and a professional network of experts. The federal government has much of the expertise necessary for a tool like LCA residing in different agencies and serving the missions of their respective departments. Sharing the expertise across agencies improves the quality of the LCA products the government produces and makes more efficient use of taxpayer funds.

The LCA Commons effort to coordinate the federal capacity has been underway in earnest since 2014. In 2012, the USDA National Agricultural Library (NAL) began the LCA Commons activity with the intention of making agricultural data and LCA related information easily available to the public. NAL set up a repository using openLCA software [6] and uploaded LCA models of commodity crop production. NAL also began hosting the USLCI for NREL using the same openLCA application on the NAL servers. This initial coordination between DOE and NAL led to a meeting hosted at the USDA headquarters in Washington DC in 2014 to explore an idea long championed by the EPA—a coordinated interagency LCA initiative. Engineers and program managers from DOE National Labs, USDA, EPA, and GSA attended this meeting. The attendees all had different perspectives on how their mission areas use LCA, but shared appreciation for the value of reusable LCA datasets. They recognized reusable, open datasets are an efficient way to share the specialized expertise of and communicate the science of each agency. A vision for a distributed network of LCA researchers and methods developers sharing LCA information through a common data interface emerged from that meeting.

Since the meeting at USDA headquarters, the USDA LCA Commons expanded into a Federal LCA Commons, including participants from FHWA, NIST, US Forest Service, six DOE national labs, and Department of Defense. In 2018, DOE, USDA, and EPA signed a three-way Memorandum of Understanding, formalizing the commitment to coordinate on

the development of LCA methods, and increase the public's access to reusable federal LCA research outputs. At the time of this publication, the Federal LCA Commons contains 18 LCI data collections available from 11 participating federal organizations. The Federal LCA Commons collaboratively developed three products that advance LCA interoperability and reuse: the US Electricity Baseline life cycle inventory (LCI) [7], the Federal Elementary Flow List [8], and the Federal LCA Commons data portal at www.lcacommons.gov.

The LCA Commons community-of-practice is currently a decentralized initiative, with independent research and policy organizations communicating to develop LCA resources of common interest. NAL currently provides repository services for discovery and access of LCA Commons data products and provides metadata guidance and support. Each participant in the community of practice shares their perspective on the suitable and necessary applications of LCA, and how LCA data products support policy and decision making. The community develops working principles for deciding on priorities for the initiative through open communication and decides how to leverage the existing mission and funding to advance the federal LCA capacity as a whole.

## 2. Challenges with Sharing LCA Data

Sharing LCA expertise between agencies through the publication, exchange, and reuse of LCI data is the primary driver for the LCA Commons. Currently, agencies can develop LCA models with deep specificity in areas of mission alignment. For example, the National Energy Technology Lab has experience developing fossil fuel pathways and carbon dioxide capture, storage, and utilization technologies. Argonne National Lab specializes in transportation, and US EPA collects emissions data and has expertise in impact modeling. Connecting LCA models developed in a distributed way into an integrated dataset brings the government's broad expertise together into a form usable for decision support. However, combining distributed datasets created for different applications into a single framework requires aligning basic LCA concepts and definitions, as well as assuring specific data goal and scope requirements are compatible.

Firstly, no consensus definitions currently exist for the terms 'data', 'LCA data', or 'LCI data' within the LCA community. Practitioners commonly use the term 'LCA data' for complex information products representing quantitative information and a wide range of metadata describing the preparation of the data, and the goal and scope of the study the data were developed to support. For this discussion paper, we define 'LCI data' as: the information necessary to reproduce an LCI. This definition comes from the interest of the LCA Commons community to correctly reuse the models that produce LCI results as a source of secondary information for LCA. An LCI typically represents industrial or economic activities at the process or firm level. They include information products that take measured data from facilities and create a materials balance at a sufficient level of aggregation for an LCA study. This aggregation involves the application of engineering principles, LCA-specific value choices and assumptions, and direct input from subject matter experts. No specification exists for the necessary and sufficient information to reproduce an LCI, or method to determine an LCI data's fitness-for-purpose for new applications beyond its original intention. Increasing demand of reusing LCA results for new studies, and for the emergence of open data as a best practice for the scientific and information communities shines a light on this lack of specification.

The LCA community made several efforts to formalize the information necessary to support data sharing and reuse. These efforts include the EcoSpold format developed by EcoInvent [9], the ILCD format developed by the European Commission Joint Research Council [10], and the openLCA JSON-LD data model developed by GreenDelta [11]. These three formalisms all start with the ISO 14048 standard for documenting LCA process data, and then develop their own approach to capture the additional information necessary for sharing LCI data models.

The ISO 14000 series of standards underpins LCA, and ISO 14048 describes the minimum necessary meta-information for reporting individual process data. ISO14048's stated

intention is to facilitate the reporting and review of LCA information and supports information exchange, with the expectation that LCA quality and reliability review depends on process information. ISO 14048 was not expressly intended to support the interpretation of LCI information for reuse for new applications.

ISO 14048 requires documentation for a general description of the process, the sources of quantitative inputs into an LCI model, information on data collection, aggregation, and cut-off criteria, and how well a particular model supports the study's stated goal. ISO also requires other types of reporting metadata, such as identifying third party reviewers and sources of project funding. Although the LCA community considers the unit process as the foundational component of LCA data, the ISO specification does not require enough information to make it reusable. Details such as intended application, upstream input providers, intended impact methods, scope, etc., are typically insufficiently described in a single unit process metadata. As a consequence, an LCI typically cannot be recreated from a unit process or collection of unit processes without providing additional information to a modeler. This additional information includes (but is not limited to) how processes exchange inputs, how co-products are managed, which impacts the process is designed for, and the algorithm for calculating results.

Data vendors developing ISO compliant exchange formats take different approaches to adding this necessary information, which creates inconsistencies between formats and information models. The open source LCA modeling software openLCA produces a JSON-based serialization of its native data structure that is fully ISO 14048 compliant with extended metadata and documentation fields. The Federal LCA Commons adopted the openLCA data model for managing and publishing LCA data for three reasons:

1.  The data model includes a comprehensive representation of ISO 14048,
2.  OpenLCA json supports associating collections of unit processes into product systems which includes modeling decisions significant for re-use, and
3.  OpenLCA is a free open source modeling tool, enabling researchers to use the same data model and software for LCA modeling and data documentation and publication.

OpenLCA serializes its data modeling into a relational, hierarchical JSON-LD format. Independent classes of objects receive unique identifiers and full descriptions with formatted metadata and attributes. OpenLCA references instances of independent objects by their IDs. For example, openLCA treats a process as an independent object. This class of independent object has process specific metadata (as described above), and includes references to other independent objects, such as flows (describing inputs and outputs), units of measure, people, and source documents. Flows as independent objects also contain references to material such as units of measure and source information. Other data formats use this hierarchical approach as well, most notably ILCD and ESv2. However, the complexity of these relationships makes it very difficult to generate these files without the help of software designed for the task.

## 3. The LCA Commons Approach to Distributed Data Development

Deciding how to combine different models produced by different people at different agencies for different purposes into a single consistent data product remains a long-standing challenge for a distributed network of data developers such as the LCA Commons. The heart of the challenge is meeting two conditions for LCA data interoperability: distributed models are suitable for use together, and the information system can recognize and resolve the connections between them. The first condition, suitability for use, is most effectively assured with close coordination between parties during the goal and scope development of an LCA before data collection and modeling begins. The second condition, an information system to resolve connections, requires all parties adopt consistent technologies and standards.

The details and challenges of connecting distributed datasets is illustrated by the LCA concept of the product system. A product system is a set of processes or subsystems, connected to each other through the exchanges of valuable materials, energy, or services.

Each process or subsystem represents an LCA model, and usually represents a specific economic activity or production process. A complete cradle-to-grave LCA will have a product system including the full production supply chain, use, and end of life. It is these different components of a product system that are almost always developed by different parties with specific expertise. Product systems in LCA are often modeled as a graph, with a set of nodes representing processes or activities, connected by edges representing the exchange of valuable material or energy Figure 2. For the LCA Commons, the connection between processes happens through the flow object. The flow represents a specific quality of material or energy that can be exchanged between processes or firms. Examples of flows include: Electricity, AC, 120V, and Containerboard, 100% recycled, at mill. A set of separate, but related, processes combines to form a single model or dataset, perhaps within a single industry, connected by the exchanges of valuable flows. Purchased, sold, imported, or exported material and energy leaves the system boundary of a specific industry dataset, and can connect to the providers and consumers of these material and energy flows. The ability to connect these material energies depends on consistent quality of the flows between products and demands of the connected process models.

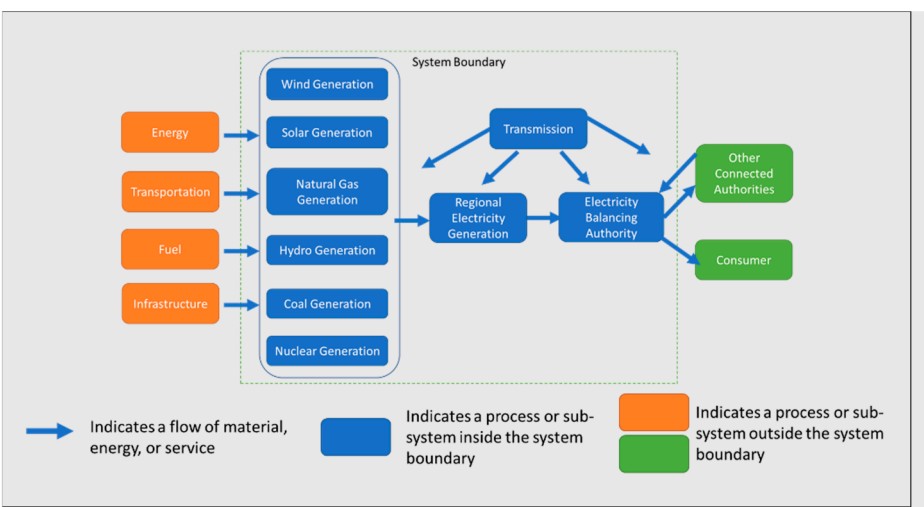

**Figure 2.** A simplified product system graph with inputs and outputs to external systems.

The graph of connected processes can extend to the industry specific models produced by different organizations working within the LCA Commons. Different classes of activities produced by different participant organizations within the LCA Commons include electricity, fuels, transportation, and waste management. Producing a reasonably accurate LCA model requires a combination of suitable independently developed datasets with consistent scopes, goal definitions, and data collection requirements. This fitness-for-purpose of a dataset as a component of a larger distributed model is either assured in advance through active coordination between the research teams developing the datasets or must be determined after the development by reviewing a dataset's metadata and documentation, and if possible, interviewing the data developers. The LCA Commons engages in both types of data interoperability activities, coordinating to develop background datasets that can interoperate between each other and serve as a background data product for a range of applications. The LCA Commons also documents and formats datasets with high reuse potential to accurately determine so their fitness-for-purpose for new applications may be accurately determined in the future.

An information system that can recognize and resolve flows is the second condition for connecting decentralized LCA models. The LCA Commons uses common flows as the method to connect different process models. For example, a process that uses electricity may have Electricity, AC, 120V as an input. OpenLCA can link a power generation process that has the flow Electricity, AC, 120V as an output to the downstream process as a provider of

that electricity flow. Datasets produced by a distributed network of research organizations can link their data together through the specification of flows.

The LCA Commons manages LCA data and information through two integrated software applications: the openLCA desktop modeling software, and the LCA Collaboration Server (CS) web repository (GreenDelta developed both applications are developed by). OpenLCA can perform LCA studies, provides the interface for creating and editing LCA data and models, and manages the information architecture to facilitate data reuse. OpenLCA can serialize models into a number of exchange formats, including the native openLCA JSON-LD, ILCD, and the superseded ecospold v1.

The LCA Collaboration Server provides a web repository for openLCA formatted JSON-LD files. The CS application offers search and access functionality for LCI datasets. It displays datasets in an easy-to-interpret format similar to the information organization structure in the openLCA software client. Web services perform information exchange between the openLCA desktop application and the CS repository—LCA datasets can be 'pushed' and 'pulled' to and from the CS directly from the openLCA interface with the click of a button. The web services handle native JSON-LD files, resulting in no information loss due to format conversion when moving information from the desktop client to the server.

Participants in the LCA Commons community prepare their data for publication in openLCA. OpenLCA does not provide instructions on how to complete fields provided in the data interface. The LCA Commons has compiled a guidance document with instruction on how to fill in the metadata fields. In addition to completing metadata, the LCA Commons also carries data organizational requirements. The LCA Commons recommends the use of the Federal Elementary Flow List, a standardized list of environmental flows developed by the US EPA consistent with the latest EPA impact methods. The guidelines specify conventions for naming process and flow data types based on recommendations from the ILCD data handbook. The LCA Commons also makes a clear distinction between process and flow objects. Processes consist of models representing activities that perform a function, and flows represent a specific quality of material or energy that can be exchanged between processes. The LCA Commons uses NAICS industry codes for the classification of process and material flows.

To ensure correct classification of distributed datasets that can connect to each other, data developers must also use a set of openLCA reference objects that facilitate interoperability. Ensuring consistent flows from provider and consumer processes remain of primary importance. OpenLCA assigns universally unique identifiers (UUID) to independent classes of objects such as flows, processes, and categories. These UUIDs allow the modelers and the software tools to identify specific unique objects. When assigning a flow intended to connect to a distributed dataset, a researcher must use the correct flow and ID, and verify openLCA will successfully make the connection. The user must perform this same activity with every shared reference object: units, flows, people, geographical locations, and NAICS classification codes. The LCA Commons maintains a template of commonly used reference objects to simplify this part of the data curation task.

The direct coordination and integrated publication workflow facilitates the building of LCA data products from reusable, distributed LCIA datasets. However, the published data are not curated data products ready to use. The complexity of LCA models, and the decentralized nature of the LCA Commons, requires that a data user build the LCA product themselves, review the documentation for suitability for a specific application, and verify that the models provide reasonable results. Without a central authority developing and maintaining a curated federal LCA data product, the responsibility of the maintenance of specific LCI data models rests with the organization that produced them. To support this distributed maintenance approach, the LCA Commons provides access to the LCI datasets as separate repositories with datasets that reference each other through common flows, but do not constitute a single, comprehensive LCA data product.

### 4. Discussion

Several lessons from developing a community of practice around sharing data and information, and building a workflow to support the community, informs the LCA Commons approach. First, when preparing LCI data for reuse, the data developer must directly contribute to documenting and formatting the data. A person not intimately familiar with how and why the data were developed cannot perform the work to a sufficient level. Second, the LCA modeling tools used to perform the original assessments must be integrated into whatever tools and workflows are necessary to prepare data for access and reuse. Third, data curation for sharing and reuse is an expensive investment. Having a clear use case and business model to support the effort is important.

NAL has performed some 'third party' data curation work, transforming LCI information from spreadsheets and reports into openLCA ready LCI datasets. This curation required working intimately with the developers of the data to supply NAL with sufficient information to conform to the openLCA data structure. The amount of interpersonal communication and the opportunity for introducing errors made it clear that involving an external documentor to generate metadata and complete reporting formats proved highly inefficient and expensive. The researcher developing the models must document process-specific information for an LCA. The contexts for the data development result in specific documentation and metainformation for LCI information. Typically, only the data developer knows the information regarding the sources for underlying observational data, how the observations were processed and aggregated, and information on assumptions and cut-offs. The data developer or the project manager knows why this data was developed, what scope of study it was developed for, and how the final data products were combined and applied to a specific research question. A person not involved in the development of the data and the LCA for which the data was developed cannot produce sufficient metadata and information due to this high level of detail for data verification and reporting. In fact, the researchers themselves may not even be able to provide enough information to engender re-use if they do not make data publication a priority during goal and scope definition. The researcher must see data publication and re-use as worthy enough an objective to ensure the models are well documented from the beginning.

The LCA information model presents complexities, with important information contained in the relationships between data objects, as well as in the documentation. Data reuse and interoperability depend on the references between processes and datasets in addition to the scope and representation of individual datasets. Stand-alone data curation and documentation tools make re-creating and verifying this relational information difficult. NAL engaged in several efforts to produce data exchange files in ILCD xml, EcoSpold V2, and JSON-LD using spreadsheets and separate desktop applications. The learning curve for using these separate tools added inefficiency without significant benefit, and more critically, using these tools divorced the data from their intended application as datasets in LCA models. Integrating data curation and preparation into openLCA gave researchers a way to prepare data for publication without learning a new tool, and allowed important insight into how the data would appear to others reusing the information. Furthermore, using webservices to transfer the data between the researcher and the remote repository increased efficiency, reduced data loss, and allowed NAL to incorporate other valuable tools, such as version control and publication workflows, directly into the modeling software [12].

Often, the best time to prepare data for reuse is when the data is being originally developed. This is when the primary engineers or scientists are actively engaged in preparing the data for the LCA study. Adding additional metadata and spending time formatting LCI for reuse during the LCA study adds time to the overall project, but it is a small investment of effort compared to performing this curation after the project is completed. After completion, the LCA specialists most familiar with the work are often assigned to new projects, and there may be little or no funding to perform data curation after a project has been completed. Most of the information available on the LCA Commons repository at www.lcacommons.gov is 'new' data that has been developed from

the beginning as data to be published on LCA Commons. In some cases, data is developed in other LCA modeling software such as SimaPro, which does not directly support the LCA Commons conventions, and is then curated in openLCA for publication by the team developing the data. The considerable expense of re-engineering LCA data for reuse after a dataset has been developed is why some federal LCA data is not immediately available on the lcacommons.gov repository despite active engagement and collaboration by the agencies who have developed it.

## 5. Conclusions

The LCA community often talks about LCA data as a type of scientific or empirical data, like economic data, surveys, or meteorological observations. However, in practice, LCA data are highly engineered information products developed for a specific application. Resulting LCI data function more as models than data, though they may be based on empirical data, engineering principles or models, and the anecdotal opinions of subject matter experts. The high value LCI data made up of information products with significant reuse potential and cost savings that underlay an LCA prove closer in form to open-source software than to empirical datasets. LCI data intended for reuse do not often 'stand alone', but are instead expected to be combined with LCI data from other sources or organizations to create a complete LCI. The complexities of the model and the expectation for reuse makes sharing LCI data more akin to sharing code through open source code repositories.

The LCA Commons developed and adopted some successful approaches with open source code development, including integration of workflows into modeling software and developing a community of practice around a common function. However, the LCA Commons, and the LCA community at large, have a long way to go before realizing some of the efficiencies of the software community. Specifications for what constitutes LCI data reuse, standard practices to document data and workflows, and consensus on best practices and exchange formats to allow for seamless interoperability remain for future work in the LCA field.

## 6. USDA EEO/Non-Discrimination Statement

The U.S. Department of Agriculture (USDA) prohibits discrimination in all its programs and activities on the basis of race, color, national origin, age, disability, and where applicable, sex, marital status, familial status, parental status, religion, sexual orientation, genetic information, political beliefs, reprisal, or because all or part of an individual's income is derived from any public assistance program. Not all prohibited bases apply to all programs. Persons with disabilities who require alternative means for communication of program information (Braille, large print, audiotape, etc.) should contact the USDA's TARGET Center at (202) 720-2600 (voice and TDD). To file a complaint of discrimination, write to USDA, Director, Office of Civil Rights, 1400 Independence Avenue, S.W., Washington, D.C. 20250-9410, or call (800) 795-3272 (voice) or (202) 720-6382 (TDD). USDA is an equal opportunity provider and employer.

**Author Contributions:** Conceptualization, E.K., E.A. and P.A.; writing—original draft preparation, E.K.; writing—review and editing, E.K., E.A. and P.A.; supervision, P.A.; project administration, E.K. and P.A. All authors have read and agreed to the published version of the manuscript.

**Funding:** This research received no external funding.

**Institutional Review Board Statement:** Not applicable.

**Informed Consent Statement:** Not applicable.

**Acknowledgments:** This work was supported by the U.S. Department of Agriculture, Agricultural Research Service.

**Conflicts of Interest:** The authors declare no conflict of interest.

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
