# Peer review of "The LCA Commons—How an Open-Source Repository for US Federal Life Cycle Assessment (LCA) Data Products Advances Inter-Agency Coordination"

_applsci, doi:10.3390/app12020865_

Round 1

Reviewer 1 Report

This is a great piece of communication article detailing the development of the Federal LCA Commons. It highlights the challenges in creating a harmonized database from various LCA data sources, which originated from various US government-sponsor programs. This work will encourage the advancement of harmonized open LCA data repository. Therefore, I strongly recommend the publication.

A few minor comments:

  1. I find the Introduction writing a bit too long. The section includes a. overview and purpose of the paper (line 24 – 47), b. general LCA introduction (line 48 – 125), which is a bit redundant for LCA practitioner but can be useful for unfamiliar audiences, and c. LCA development in the US (line 126 – 209). I would suggest adding a subsection title in between the three components to improve the reader friendliness.

  2. Format of references/citation should be improved.

  3. I am wondering what happened to some organizations in which their LCI data were not reflected in the common repository. Specifically, you mentioned quite a bit about DOE Argonne National Lab and their GREET model/database. However, the LCI items in GREET such as lithium-ion battery are currently not included. Are there any specific challenges worth discussing concerning here?

That’s all.
Thank you!

Author Response

Thank you for your helpful comments.

We have broken up the introduction into sub sections as recommended.  Hopefully this will improve the readability of the introduction section.

We have gone through and formatted the references for consistency.

We have added an additional paragraph on the end of the discussion that addresses why some actively engaged members of the community, with popular data products, do not have their data on the lcacommons.gov repository. This question from the reviewer brings up an important consideration around data curation that was not originally addressed.

Reviewer 2 Report

The research theme of the paper "The LCA Commons – an open source repository for US federal Life Cycle Assessment data products" is relevant and interesting, but for its acceptance the authors need to make major adjustments.
The title is not clear enough, in addition to using acronyms, which although obvious to readers of the topic should be avoided.
The summary is not clear, it lacks several fundamental information for its reading and attractiveness;
The manuscript as a whole does not have the characteristics of a scientific paper, it lacks in-depth discussions and adequate references. This paper as it stands cannot be accepted, it must be profoundly improved, it must look like a scientific article!!! as it is, it is not possible with all due respect to the authors. The topic is of interest to readers, but these corrections are essential.

Author Response

Thank you for your comments.

We have elaborated on the title of the paper, including explicitly defining the acronym LCA.  The paper title is the name of a government program, LCA Commons, so in this case we believe the acronym is appropriate.

It would be helpful if the reviewer could elaborate on which fundamental information is missing from the abstract.

We accept that this paper does not follow the format of scientific research submissions.  This work does not reflect novel research, it is a specific government program at the USDA that advances open data.  The paper was constructed to reflect as such.  If there are specific citations missing the reviewer is aware of we would appreciate some hint as to what they are. Similarly, if there are specific areas of discussion that could be expanded that would add value to the overall message, please let us know.

This paper was submitted as a commentary after we were invited by Wes Ingwersen at US EPA to submit a paper on how this particular government program is contributing to open data.  Perhaps this work is not appropriate for this publication after all.

Round 2

Reviewer 2 Report

Ok.